# Hybrid Structures of Sisal Fiber Derived Interconnected Carbon Nanosheets/MoS_2_/Polyaniline as Advanced Electrode Materials in Lithium-Ion Batteries

**DOI:** 10.3390/molecules26123710

**Published:** 2021-06-18

**Authors:** Wei Li, Yuanzhou Liu, Shuang Zheng, Guobin Hu, Kaiyou Zhang, Yuan Luo, Aimiao Qin

**Affiliations:** Key Lab New Processing Technology for Nonferrous Metals & Materials Ministry of Education, Guangxi Key Laboratory in Universities of Clean Metallurgy and Comprehensive Utilization for Non-ferrous Metals Resources, College of Materials Science & Engineering, Guilin University of Technology, Guilin 541004, China; liwei1986gllg@163.com (W.L.); yuanzhou_liu@126.com (Y.L.); zhengshuang981219@126.com (S.Z.); kokpinhu@hotmail.com (G.H.); kaiyou2005@163.com (K.Z.); Luo1500935739@outlook.com (Y.L.)

**Keywords:** interconnected carbon nanosheets, MoS_2_, polyaniline, hybrid architecture, lithium-ion battery

## Abstract

In this work, we designed and successfully synthesized an interconnected carbon nanosheet/MoS_2_/polyaniline hybrid (ICN/MoS_2_/PANI) by combining the hydrothermal method and in situ chemical oxidative polymerization. The as-synthesized ICNs/MoS_2_/PANI hybrid showed a “caramel treat-like” architecture in which the sisal fiber derived ICNs were used as hosts to grow “follower-like” MoS_2_ nanostructures, and the PANI film was controllably grown on the surface of ICNs and MoS_2_. As a LIBs anode material, the ICN/MoS_2_/PANI electrode possesses excellent cycling performance, superior rate capability, and high reversible capacity. The reversible capacity retains 583 mA h/g after 400 cycles at a high current density of 2 A/g. The standout electrochemical performance of the ICN/MoS_2_/PANI electrode can be attributed to the synergistic effects of ICNs, MoS_2_ nanostructures, and PANI. The ICN framework can buffer the volume change of MoS_2_, facilitate electron transfer, and supply more lithium inset sites. The MoS_2_ nanostructures provide superior rate capability and reversible capacity, and the PANI coating can further buffer the volume change and facilitate electron transfer.

## 1. Introduction

With the increasing use of fossil energy, more and more harmful gases such as SO_2_, NO_×_, and CO have billowed into the atmosphere, and the energy and environmental problems are increasingly severe. In this century, solving the serious energy and environmental problems is one of the biggest challenges facing the scientific community. In the past 10 years, multiform new energy has had rapid development such as solar energy [1], wind energy [2], tidal energy [3], and so on. However, these new energies are interval energy, which need to be stored for use. For rural areas and the community, using accumulators such as a solar and generating electricity system to store energy and building a decentralized energy system are the best solutions [4]. Meanwhile, developing electric vehicles (EVs) is also an effective measure to cope with the energy and environmental crises. However, developing a high power and energy accumulator is the key to developing the decentralized energy system and the EVs. Lithium-ion batteries (LIBs), as a high–efficiency accumulator, have been applied in a variety of fields from portable electronics to electric transportation, since their invention in 1991. However, the low capacities of commercial anode materials like graphite (372 mAh/g) limit the LIBs used in large energy storage equipment and EVs. Thus, there is an urgent need to develop high energy and power anode materials for LIBs.

Biomass derived carbon materials, especially biomass derived carbon nanosheets, usually have rich pore structure, large specific surface, and good electrical conductivity, which make them show high capacity, superior rate performance, and cycling stability when applied as anode materials for LIBs. Meanwhile, biomass is a renewable resource, has the advantages of being environmentally friendly, low cost, and easy processing, which are not available in fossil materials. In recent years, various biomass derived carbons have been employed to LIBs anode materials such as rice husk [5], peanut shells [6], and banana fibers [7]. Nevertheless, despite these marvelous features, biomass derived carbon materials as anodes for LIBs is hampered by distinct hysteresis in the voltage profile and the irreversible capabilities [8].

MoS_2_ is a typical two-dimensional (2D) transition metal sulfide consisting of a hexagonally packed layer of Mo atoms sandwiched between two layers of S atoms, and the triple layers are stacked and held together by van der Waals interactions [9]. The weak van der Waals interactions can offer a convenient environment between the adjacent interlayer for reversibly inserting and extracting a mass of lithium ions. Moreover, MoS_2_ can be converted into the metal Mo and LiS_2_ with additional lithium storage capacity; therefore, MoS_2_ has a high lithium storage capacity (ca.670 mAh/g with 4 mol of Li^+^ insertion per formula) and is regarded as a promising anode material for LIBs [10,11,12]. However, MoS_2_ with a single layer and/or few layers are easy to stack and restack, and combined with the inherent poor electrical conductivity and the conspicuous volume change, greatly inhibit the potential of MoS_2_ for use in LIBs.

Polyaniline (PANI), as a common conducting polymer, has been found to be the most promising because of its easy to synthesis, low cost, tunable properties and good stability [13]. Utilizing the advantage of PANI to embellish the surface of anode materials is an important way to improve the electrochemical performance of anodes. The PANI coating on the surface of anode materials can form the core-shell structures, which was proven to be particularly advantageous. First, the core-shell electrode can achieve synergetic improvements from the intrinsic properties of each component such as better electrical conductivity, shorter ionic transport, improved mechanical stability, and cycling stability [14]. Second, the unique coating can decline the surface energy effectively, which will reduce the aggregation possibilities of active materials and relieve side reactions between the electrolyte and electrode, bringing about better reversibility and cycling stability of the electrodes [15,16]. Zheng [17] successfully prepared TiO_2_/PANI composites using a solid coating method, and the composites showed an increased reversible capacity compared with pure TiO_2_. At the current density of 20 and 200 mA/g, the maximum capacities of TiO_2_/PANI were 281 and 168.2 mAh/g, respectively, while the pure TiO_2_ was only 230 and 99.6 mAh/g, indicating that the PANI coating can improve the capacity and rate capability. Liu [18] synthesized a tremella-like hierarchical porous MoS_2_/PANI composite anode via a facile polymerization and hydrothermal method, which offered a high reversible capacity of 915 mAh/g at the current density of 1.0 A/g after cycling 200 cycles, while the pure MoS_2_ anode decayed severely to about 42 mAh/g, suggesting that the PANI coating can significantly enhance the cycle stability.

Herein, considering the advantages of biomass derived carbon nanosheets, MoS_2_, and PANI coating, we designed and successfully fabricated a caramel treat-like ICN/MoS_2_/PANI hybrid via a facile hydrothermal method and polymerization for LIB application. It was found that the ICN as the framework can not only facilitate the decentralization and buffer the volume change of MoS_2_, but can also facilitate electron migration and supply more lithium inset sites. The MoS_2_ can provide superior rate capability and high reversible capacity, and the PANI coating can further buffer the volume change of MoS_2_ and facilitate electron migration. Thus, every component in the hybrid can complement each other and excellent electrochemical performance can be achieved.

## 2. Results and Discussion

The morphology of the as-prepared ICN/MoS_2_ and ICN/MoS_2_/PANI hybrid was characterized by SEM (Figure 1). Figure 1a shows that the MoS_2_ grew in a cluster on the surface of the meso-microporous ultrathin 3D ICNs with a slice thickness of about 2 nm (Appendix A, Appendix A). High magnification (Figure 1b) of the MoS_2_ clusters showed that they had a flower-like nanostructure with diameter of ~200 nm and lamellar thickness of ~2 nm. In Figure 1c, the ICN/MoS_2_/PANI presents a “caramel treat-like” architecture, and the high magnification SEM image (Figure 1d) revealed that the PANI was evenly coated on the surface of the ICNs and MoS_2_.

The detailed structure of the as-prepared ICN/MoS_2_/PANI hybrid was further characterized by TEM (Figure 2a,b). Figure 2a clearly reveals the typical “caramel treat-like” structure of ICN/MoS_2_/PANI, and the transparency implies the coated PANI layer. The high-resolution TEM image demonstrated the edge view of a dark spot with some visible lattice fringes (Figure 2b). The interplanar distances were carefully measured to be about 0.18, 0.22, and 0.27 nm, corresponding to the interlayer spacing of (105), (103), and (100) planes of 2H-MoS_2_, respectively. FESEM element mapping was also implemented to further confirm the element distribution in the ICN/MoS_2_/PANI hybrid. Figure 2c shows the location of ICN/MoS_2_/PANI for EDS-mapping. As shown in Figure 2d–f, the Mo, S, and N elements were uniformly distributed over the whole area of the ICN/MoS_2_/PANI hybrid, indicating a homogeneous distribution of MoS_2_ and PANI in the hybrid.

X-ray powder diffraction (XRD) was carried out to investigate the crystallographic structure of the as-prepared ICN/MoS_2_, PANI and ICN/MoS_2_/PANI, as shown in Figure 3a. The diffraction peaks for both ICNs/MoS_2_ and ICNs/MoS_2_/PANI at the 2θ values of 14.12°, 32.19°, 39.51°, 49.41°, and 58.76° can be respectively indexed as (002), (100), (103), (105), and (110) crystal of hexagonal MoS_2_ with molybdenite 2H crystalline structure (JCPDS card no. 75-1539), and the broad peak at around 24° can be attributed to the (002) plane of ICNs. For pure PANI, three crystalline peaks were observed at 15.1°, 20.5°, and 25.5° corresponding to (011), (020), and (200) planes of PANI, indicating that the PANI existed in emeraldine salt form [19,20]. While for the ICN/MoS_2_/PANI hybrid, there were three similar crystalline peaks at 15.1°, 20.5°, and 25.5° to that of pure PANI, implying that PANI existed in this hybrid material, matching well with the SEM and TEM images. The ICN/MoS_2_/PANI hybrid was further confirmed by FTIR spectra (Figure 3b). It can be clearly seen that the ICN/MoS_2_/PANI exhibited both characteristic bands of PANI at 1559, 1480, 1293, 1239, 1106, 794 cm^−1^, and ICN/MoS_2_ at 1591, 1400, 1123, and 590 cm^−1^, revealing its hybrid structure. Among these bands, 1559 and 1480 cm^−1^ were associated with the C=C stretching mode for the quinoid (Q) and benzenoid (B) rings, respectively [21]; 1293 and 1239 cm^−1^ belonged to the C–N stretching of an aromatic amine; 1106 and 794 cm^−1^ were attributed to the in-plane and out-plane bending vibration of C–H in Q and B rings, respectively; and the strong band represents the good conductivity of PANI in the hybrid [22,23].

To research the porous features, the nitrogen adsorption/desorption isotherms and pore size distribution (PSD), as calculated by the QSDFT model of ICN/MoS_2_ and ICN/MoS_2_/PANI, are exhibited in Figure 4. In Figure 4a, the type І/IV isotherm with H3 hysteresis (according to IUPAC classification) loops was observed for ICN/MoS_2_ and ICN/MoS_2_/PANI, indicating that they both take on open slit-like meso-microporous and platelet structures [24,25]. These results can be further confirmed by the PSD (Figure 4b), where it can be seen that ICN/MoS_2_ and ICN/MoS_2_/PANI mainly consist of micropores with a pore diameter <2 nm, and contain some mesopores with a porous diameter of 2~5 nm. Moreover, ICN/MoS_2_/PANI had fewer pores than ICN/MoS_2_, which was due to the PANI coating on the surface. The BET specific surface areas of ICN/MoS_2_ and ICN/MoS_2_/PANI were 518.9 and 36.4 m^2^/g, respectively, sharply reducing the specific surface areas of ICN/MoS_2_/PANI, which may be that part of the pores of the material was blocked by the PANI coating.

The lithium storage mechanism of the ICN/MoS_2_/PANI was investigated by cyclic voltammogram (CV) and galvanostatic charge–discharge curves. Figure 5a,b, Appendix A (Appendix A) show the initial three cycles CV cures of the ICN/MoS_2_/PANI, ICN/MoS_2_, ICNs/PANI, and pure PANI at a scan rate of 0.2 mV/s, respectively. As shown in Figure 5a, during the first cycle, the ICN/MoS_2_/PANI hybrid displayed three reduction (Li insertion) peaks at 1.07, 0.49, and 0.02 V, and two oxidation (Li extraction) peaks at 1.62 and 2.32 V. The reduction peak at 1.07 V can be assigned to Li insertion into the interlayer of MoS_2_, accompanied by phase transformation from 2H to 1T structure of Li_χ_MoS_2_ [10]. The pronounced reduction peak at 0.49 V can be ascribed to the reduction of Li_χ_MoS_2_ to Mo metal and Li_2_S. The spiculate reduction peak at 0.02 V can be attributed to Li insertion into the interlayer of carbon (ICNs). The oxidation peak at 1.62 V can be ascribed to partial oxidation of Mo to form MoS_2_, and the following oxidation peak located at 2.32 V was associated with the oxidation of Li_2_S into S [26]. In the second cycle, the peaks at 1.07 and 0.49 V disappeared, and two new peaks at about 2.0 and 1.3 V appeared for the conversion of S to Li_2_S and the association of Li with Mo [27,28], respectively. In the third cycle, the CV curve was almost coincident with the CV curve of the second cycle, indicating that the redox pair was highly reversible. For the ICN/MoS_2_, in the first cycle, the reduction and oxidation peaks appeared at 1.08/0.46/0.02 V, and 1.61/2.32 V; in the second cycle, the peaks at 1.08/0.46 V disappeared and two new peaks at around 2.0/1.3 V appeared; and in the third cycle, the CV curve was almost coincident with the second cycle CV curve: all these were quite similar to the ICN/MoS_2_/PANI hybrid. Comparing the CV curves of ICN/MoS_2_/PANI with ICN/MoS_2_, ICN/PANI and PANI, it can be seen that the redox current of ICN/MoS_2_/PANI was higher than that of ICN/MoS_2_, ICN/PANI, and PANI, which demonstrates that the ICN/MoS_2_/PANI hybrid showed faster kinetics and higher capacity [29], so the coating of PANI on ICN/MoS_2_ can not only provide more pathways for electric transfer, but also alleviate the volume change during cycling, thus a better electrochemical performance can be achieved.

Galvanostatic charge–discharge curves of the ICN/MoS_2_/PANI and ICN/MoS_2_ anodes are shown in Figure 5c,d, respectively, which conforms to the aforementioned CV curves. The initial discharge (Li insertion) capacity of the ICN/MoS_2_/PANI and ICN/MoS_2_ was 1293 and 1008 mAh/g at a current density of 0.05 A/g, with the charge (Li extraction) capacities of 679 and 681 mAh/g, corresponding to the initial Coulombic efficiency of 53% and 68%, respectively. The charge/discharge capacities of the ICN/MoS_2_/PANI in the second and third cycles almost had no degradation, implying its high reversible capacities. Though the initial Coulombic efficiency of ICN/MoS_2_/PANI was lower than that of ICN/MoS_2_, its initial discharge capacity and cycle stability were enhanced. The low Coulombic efficiency could be caused for the formation of a solid electrolyte interphase (SEI) layer and any irreversible lithium insertion into this hybrid structure [30].

Galvanostatic charge–discharge technique was used to further study the cycle performance of ICN/MoS_2_/PANI. Figure 6a shows the specific discharge capacity from the 1st to 500th charge/discharge cycles at a high current density of 2.0 A/g in a voltage window from 0.01 to 3.0 V (vs. Li^+^/Li). After 400 cycles, ICN/MoS_2_/PANI and ICN/MoS_2_ anodes both demonstrated a high reversible capacity of 583 and 527 mAh/g, respectively, with a high Coulombic efficiency over 99%, while the reversible capacity of ICN/PANI and pure PANI was only 251 mAh/g and 25 mAh/g after 200 cycles (Appendix A, Appendix A), respectively. Remarkably, the ICN/MoS_2_/PANI anode exhibited a higher capacity and better cycling performance at high current density than that of ICN/MoS_2_, ICN/PANI, and pure PANI. The improved capacity could be attributed to the elastic PANI conductive polymer and the hierarchical structure. It should be noted that the capacity of the ICN/MoS_2_/PANI distinctly increased after 400 cycles. The increment of capacity should be mainly attributed to the activation process of the hierarchical structure as the flower-like MoS_2_ gradually expanded and exfoliated during cycles, which can provide more lithium storage sites and a low energy barrier for lithium intercalation or deintercalation [18]. Appendix A (Appendix A) shows the specific discharge capacity of ICN/MoS_2_/PANI in long cycles at a high current density of 2.0 A/g. After 1000 cycles, ICN/MoS_2_/PANI showed a high reversible capacity of 635 mAh/g, which further confirmed the cycling stability of ICN/MoS_2_/PANI.

Rate performances of the ICN/MoS_2_/PANI and the ICN/MoS_2_ were measured and compared. As shown in Figure 6b, the ICN/MoS_2_/PANI electrode delivered a reversible discharge capacity of about 747, 645, 582, 540, 454, 408, and 350 mAh/g when the current density was 0.05, 0.1, 0.3, 0.5, 1.0, 2.0, and 3.0 A/g, respectively. When the current density was restored to 0.05 A/g after 35 cycles at different rates, the reversible capacity could still be recovered to 711 mAh/g, indicating the highly stable cycling and superior rate performance of the ICN/MoS_2_/PANI hybrid. Regarding the ICN/MoS_2_, the reversible capacities were comparable with those of ICN/MoS_2_/PANI at the low current density of 0.1, 0.3, and 0.5 A/g, while at the high current density of 1.0, 2.0, and 3.0 A/g, the reversible capacities of ICN/MoS_2_/PANI were all higher than those of ICN/MoS_2_, and also much higher than pure MoS_2_ [31]. This may be attributed to fact that the PANI provided a highly conductive medium for electron transfer during the charge/discharge cycle. Meanwhile, the caramel treat-like structure of the ICN/MoS_2_/PANI hybrid not only enabled efficient active electrode–electrolyte contact, but also accommodated the strains related to volume changes during repeated charge/discharge cycles.

Electrochemical impedance spectra (EIS) measurements were also carried out at the frequency range from 0.1 MHz to 0.01 Hz to investigate the electron transfer mechanisms of the ICN/MoS_2_/PANI in LIBs. In Figure 6c, all of the Nyquist plots consisted of two parts: a semicircle, relating to the electrolyte resistance (Rs) and electron transfer resistance (Rct) at high frequencies, an inclined line relating to the ion diffusion (Zw), and electrochemical double-layer capacitive behavior (Cdl) at low frequency. The Nyquist plots of ICN/MoS_2_/PANI and ICN/MoS_2_ were both fitted by the appropriate electric equivalent circuit (inset of Figure 6c). According to the fitting results, the Rct value of the ICN/MoS_2_/PANI electrode was 49 Ω, while the Rct value of the ICN/MoS_2_ was 85 Ω. The Rct value of the ICN/MoS_2_/PANI was much lower than that of ICN/MoS_2_, indicating that the PANI coating can effectively facilitate the electron transfer and improve the electrochemical performance.

In order to further understand the high-rate performance, the diffusion behavior of lithium ions on ICN/MoS_2_/PANI and ICN/MoS_2_ was studied and analyzed by the EIS test. Figure 7 is a graph of the lithium ion diffusion coefficients of ICN/MoS_2_/PANI and ICN/MoS_2_ The graph used Zw as the ordinate and ω as the abscissa. The relationship between the Warburg impedance (Zw) and the angular frequency (ω) is represented as Equation (1). The slope K of the straight line in Figure 7 is the impedance factor σ. The lithium ion diffusion coefficient D_Li_^+^ can be calculated according to Equation (2), and D_Li_^+^ is inversely proportional to the impedance factor σ, that is, the smaller the slope, the larger the diffusion coefficient [32,33].
(1)$$\mathrm{Zw}=\;\sigma \omega {}^{-0.5}\;$$
(2)$$D_{{\mathrm{Li}}^{+}}=\frac{R^{2}T^{2}}{2A^{2}n^{4}F^{4}C_{\mathrm{Li}}{}^{2}\sigma^{2}}$$

In Equation (2), R is the gas constant (R = 8.314 J·mol^−1^k^−1^) and T is the absolute temperature of the experimental environment. The area of the cathode material of A is immersed in the electrolyte, and n is the Number, F is the Faraday constant, and C_Li_^+^ is the molar concentration of lithium ions.

The diagonal line in the low frequency region is due to the diffusion of lithium ions into the active material. The diffusion of lithium ions on the active material can be characterized by the diffusion coefficient. The diffusion coefficient can be calculated according to Equations (1) and (2) using the data in the low frequency region. Zw is used as the vertical axis and ω is the horizontal axis. The slope K is the impedance factor σ, which is inversely proportional to the diffusion coefficient. As shown in Figure 7, the slope of the ICN/MoS_2_/PANI straight line was smaller than that of ICN/MoS_2_ (KICNs/MoS_2_ > KICNs/MoS_2_/PANI), which means that the ICN/MoS_2_/PANI had a larger diffusion coefficient and better diffusion performance for lithium ions.

## 3. Experimental Section

All reagents were of analytical grade and used without further purification. All kinds of aqueous solutions were prepared with deionized water. 

Figure 8 is the schematic illustration of the formation of a caramel treat-like ICN/MoS_2_/PANI hybrid.

Preparation of ICNs/MoS_2_: First, the sisal fiber derived interconnected carbon nanosheets (ICNs) with porous structures were prepared by a combination of hydrothermal and activation processes (Appendix A, Appendix A). Then, the ICNs were used as a framework via a facile hydrothermal-anneal strategy to prepare the ICN/MoS_2_ hybrid. A typical reaction, 0.1 g of ICNs, 0.06 g of MoO_3_, and 0.092 g of ammonium dithiocarbamate (CH_6_N_2_S_2_) was mixed in 70 mL deionized water with stirring for 30 min at room temperature, then the obtained black suspension was transferred into a 100 mL Teflon stainless steel autoclave, and heated at 180 °C for 36 h in an oven. After being cooled down to room temperature naturally, the precipitates were collected by filtration, washed several times with ethanol and deionized water, and dried in a vacuum at 60 °C for 8 h, followed by annealing treatment in a conventional tube furnace at 800 °C for 1 h at the heating rate of 3 °C/min in a flowing N_2_ atmosphere to increase the crystallization. Then, the precursor ICN/MoS_2_ was obtained and ready for coating with PANI. 

Preparation of “caramel treat-like” ICNs/MoS_2_/PANI hybrid: This hybrid was prepared by in situ oxidative polymerization of aniline monomers on ICN/MoS_2_ using ammonium persulfate ((NH_4_)_2_S_2_O_8_, APS) as the oxidizer. Typically, 50 mg of precursor ICN/MoS_2_ and 70 mL of 1.5 M H_2_SO_4_ were added in a flask with vigorous magnetic stirring for 15 min, then 0.045 mL aniline was injected into the solution, and continuously and vigorously stirred for 2 h. Next, APS with a molar ratio of 1:0.5 (aniline:APS) was added into the solution, and reacted for another 12 h under constant stirring at room temperature. The resulting product was filtered and washed several times with deionized water and ethanol and dried in a vacuum at 60 °C for 10 h before the caramel treat-like ICN/MoS_2_/PANI hybrid was finally obtained.

Materials characterization: Field-emission electron microscopy (FESEM, S-4800, Hitachi, Japan), and transmission electron microscopy (TEM, JEM-2100F, JEOL, Tokyo, Japan) were used to observe the morphology of the obtained materials. The specimens for TEM observation were prepared by dispersing the material powder into ethanol with ultrasonic treatment. Energy dispersive spectroscopy (EDS, an accessory of FESEM) was performed to investigate the elemental mapping of the prepared materials. Powder X-ray diffraction (XRD, X’Pert PRO, PANalytical B.V.) with Cu Kα radiation (λ = 1.54056Å), and Fourier transform infrared spectroscopy (FTIR, Nexus 470, Thermo Nicolet) were used to investigate the crystallographic and phase purity of the materials. The nitrogen adsorption-desorption analysis was performed with a surface area analyzer (NOVA1200e, Quantachrome) at 77.35 K. The specific surface area (SSA) was calculated by the Brunauer–Emmett–Teller (BET) theory, and the pore size distribution (PSD) was analyzed by a quenched solid density functional theory (QSDFT) model. 

Electrochemical measurements: The electrochemical measurements were performed under room temperature with two-electrode coin half-cells (size: 2025) that use lithium foil as the counter/reference electrode, Celgard 2400 as the separator, and 1.0 M LiPF_6_ in ethylene carbonate/dimethyl carbonate/ethyl methyl carbonate (EC/DMC/EMC, 1:1:1 *v*/*v*/*v*) as the electrolyte. The working electrodes were prepared by mixing 80 wt% active materials (e.g., as-prepared ICN/MoS_2_/PANI hybrid), 10 wt% conductivity agents (acetylene black), and 10 wt% binder (polyvinylidene difluoride, PVDF, in N-methyl-2-pyrrolidone), and ground to a homogeneous slurry. The slurry was coated on a 10 μm thick Cu foil, and dried at 120 °C for 10 h in a vacuum oven. The coin-type cells were assembled in an argon-filled glovebox with both the moisture and the oxygen content below 1 ppm. The galvanostatic charge/discharge measurements were performed with a Neware battery tester over a voltage window from 0.01 to 3 V. The cyclic voltage (CV) test was conducted using a CHI-760D electrochemical workstation at 0.2 mV/s over the potential window range of 0.01~3.0 V. The electrochemical impedance spectroscopy (EIS) was carried out on a CHI-760D electrochemical workstation in the frequency range from 0.1 MHz to 0.01 Hz.

## 4. Conclusions

In summary, a caramel treat-like ICN/MoS_2_/PANI hybrid was synthesized by a combined hydrothermal method and in situ chemical oxidative polymerization. This hybrid displayed a hierarchical architecture consisting of ICNs, flower-like MoS_2_, and homogeneously integrated PANI. The ICNs as the framework can not only facilitate the decentralization and buffer the volume change of MoS_2_, but can also facilitate electron transfer and provide more lithium inset sites. The MoS_2_ can provide superior rate capability and high reversible capacity, and the coated PANI can further buffer the volume change of MoS_2_ and expedite electron transfer. Thus, the unique ICN/MoS_2_/PANI hybrid exhibits standout electrochemical performance including excellent cycling performance, improved rate capability, and reversible capacity. This caramel treat-like ICN/MoS_2_/PANI hybrid delivered a high reversible capacity of 583 mAh/g after 400 cycles at 2.0 A/g, which is much higher than that of graphite, indicating its potential for LIB applications.

## Figures and Tables

**Figure 1 molecules-26-03710-f001:**
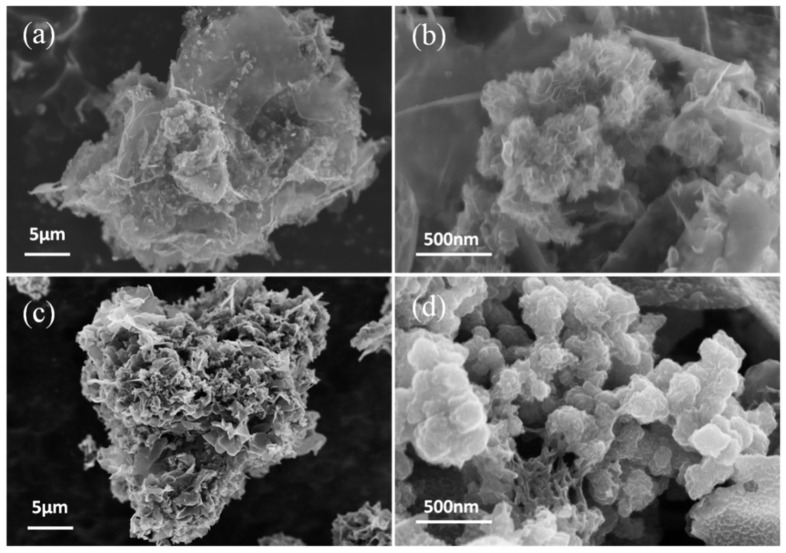
SEM images of ICN/MoS_2_ (**a**,**b**) and the ICN/MoS_2_/PANI hybrid (**c**,**d**).

**Figure 2 molecules-26-03710-f002:**
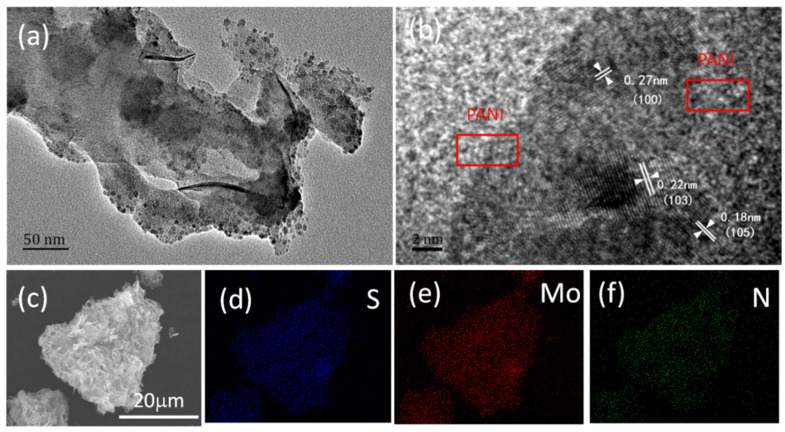
TEM images of the ICN/MoS2/PANI hybrid (**a**,**b**), SEM images of ICN/MoS2/PANI (**c**), FESEM element mapping images of ICN/MoS2/PANI hybrid for Mo (**d**), S (**e**), and N (**f**).

**Figure 3 molecules-26-03710-f003:**
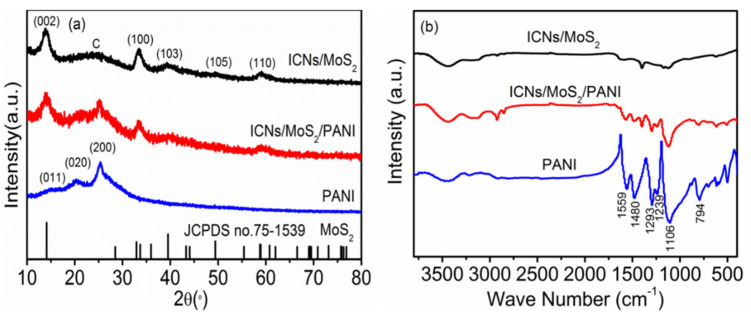
XRD patterns (**a**) and FTIR spectra (**b**) of ICN/MoS_2_, PANI and ICN/MoS_2_/PANI.

**Figure 4 molecules-26-03710-f004:**
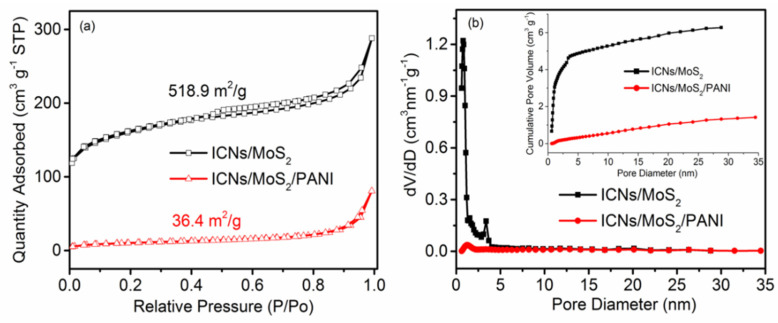
N_2_ adsorption/desorption isotherm (**a**), and pore size distribution of ICN/MoS_2_ and ICN/MoS_2_/PANI calculated by the QSFT model. (**b**) Inset: cumulative pore volume.

**Figure 5 molecules-26-03710-f005:**
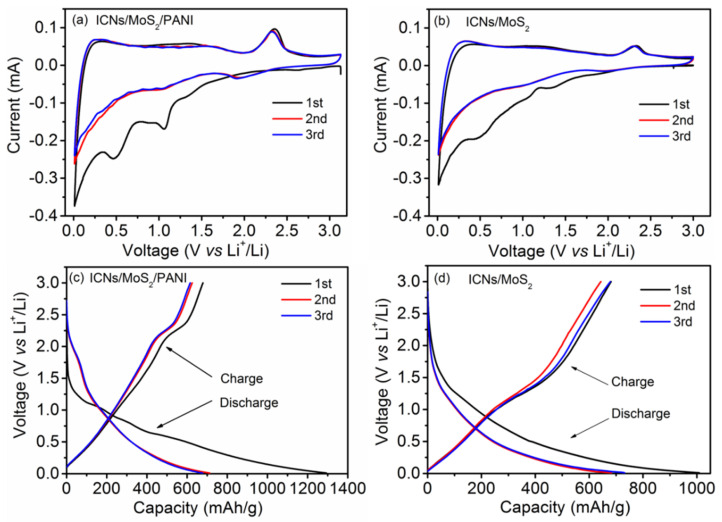
Cyclic voltammograms of the ICN/MoS_2_/PANI and ICN/MoS_2_ electrodes at a scan rate of 0.2 mV/s (**a**,**b**). Voltage-capacity plots of the ICN/MoS_2_/PANI and ICN/MoS_2_ electrodes at a current density of 0.05A/g (**c**,**d**).

**Figure 6 molecules-26-03710-f006:**
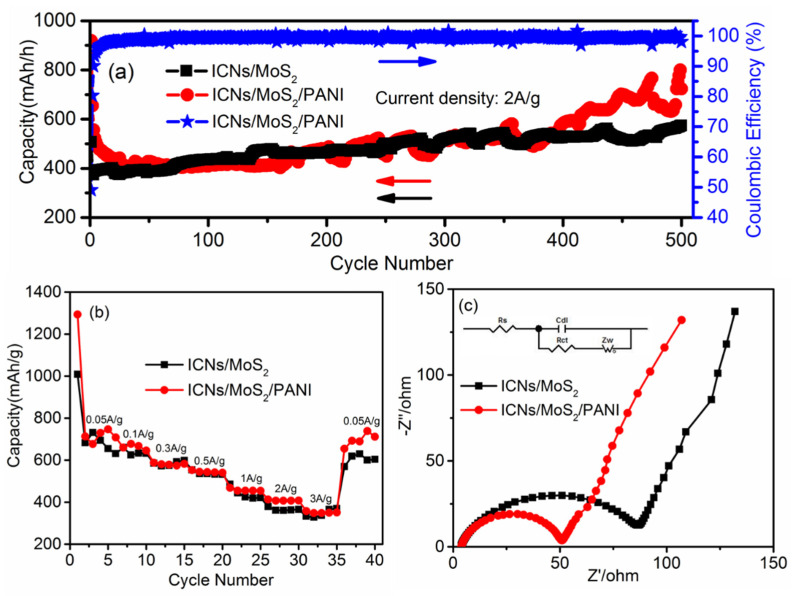
Cycling performance and Coloumbic efficiency of the ICN/MoS_2_/PANI hybrid and ICN/MoS_2_ at a current density of 2A/g for 500 cycles (**a**). Discharge capacities of the ICN/MoS_2_/PANI hybrid and ICN/MoS_2_ at various current densities (**b**). Nyquist plots of the ICN/MoS_2_/PANI hybrid and ICN/MoS_2_ (**c**). Inset: equivalent circuit model.

**Figure 7 molecules-26-03710-f007:**
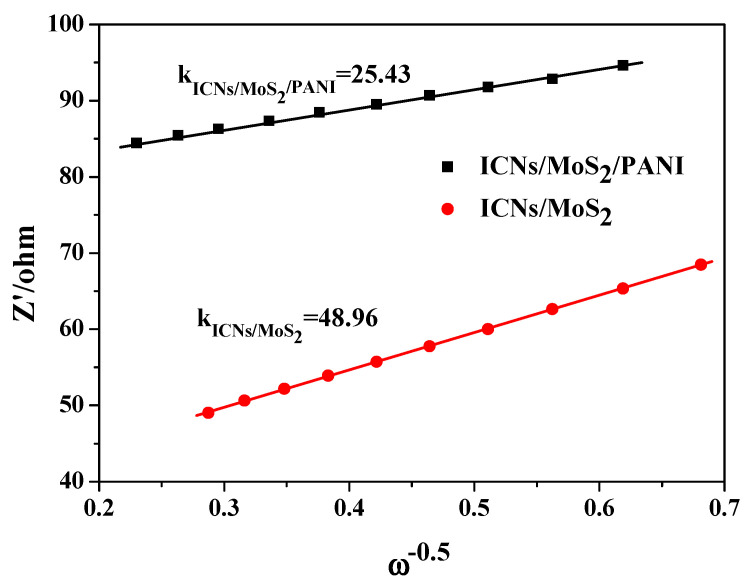
The corresponding relationship between Z′ and ω^−0.5^ of ICN/MoS_2_/PANI and ICN/MoS_2_.

**Figure 8 molecules-26-03710-f008:**
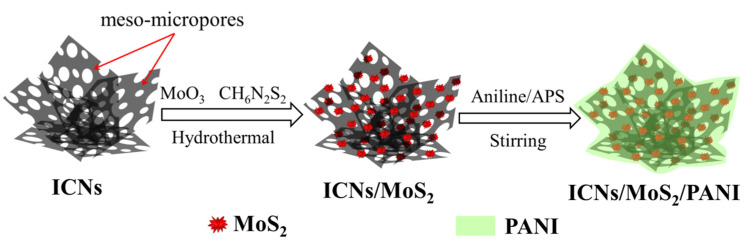
Schematic diagram of the preparation process of the ICN/MoS_2_/PANI hybrid.

## Data Availability

Data is contained within the article.

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
