# Peer review of "Hybrid Structures of Sisal Fiber Derived Interconnected Carbon Nanosheets/MoS2/Polyaniline as Advanced Electrode Materials in Lithium-Ion Batteries"

_molecules, 2021, doi:10.3390/molecules26123710_

Round 1

Reviewer 1 Report

- Within the experimental part more details need to be included such as:

          - Synthesis of “pure” PANI conditions

          - Details on the EIS measurements: at what SOC/Potential was EIS recorded? Conditions of the measurments: galvanostatic, or potentiostatic EIS, AC amplitude, DC level?

- on Page 3/ line 99: Fig.S1, Supporting information -> however I could not find a supporting information  

-In general PANI has 3 redox states with different electric conductivities. The redox behavior in aqueous electrolytes is very well understood and reported. However, in aprotic electrolytes there is much less information but it is known, that PANI is redox active and can change states and correspondingly electrical conductivity. At 3V vs. Li, PANI is in its halfoxidized emeraldine state. However, in the broad voltage window 3->0V vs. Li the emeraldine state could undergo further reduction to the leukoemeraldine state. This can be an explanation to the observed lower coulombic efficiency if the reduction is irreversible, or the increased capacity in the case the redox reaction is reversible. Therefore, I recommend that the authors perform additional CV and charge-discharge measurements on pure PANI, coated on the Cu foil and/or on ICNs/PANI.

-  The capacity of ICNs/MoS2/PANI is rising after 500 cycles. How reproducible is this effect? In general, nowhere in the manuscript standard deviations or error bars are reported. How reproducible is the observed electrochemical behavior and the cyclic performance? How many cells were tested? This is especially crucial, when discussing quantitatively the specific materials properties. The authors should include also information and comment on the reproducibility of the presented preparation procedures.

Author Response

  1. Within the experimental part more details need to be included such as: Synthesis of “pure” PANI conditions

Response: Thanks for your comments. More details of the synthesis conditions

 of “pure” PANI and ICNs/PANI have been added in the supporting information.

  1. on Page 3/ line 99: Fig.S1, Supporting information -> however I could not find a supporting information

Response: Thanks for your comments. Now, Fig.S1 in the Supporting information has been uploaded with the revised manuscript.

  1. In general PANI has 3 redox states with different electric conductivities. The redox behavior in aqueous electrolytes is very well understood and reported. However, in aprotic electrolytes there is much less information but it is known, that PANI is redox active and can change states and correspondingly electrical conductivity. At 3V vs. Li, PANI is in its half oxidized emeraldine state. However, in the broad voltage window 3->0V vs. Li the emeraldine state could undergo further reduction to the leukoemeraldine state. This can be an explanation to the observed lower coulombic efficiency if the reduction is irreversible, or the increased capacity in the case the redox reaction is reversible. Therefore, I recommend that the authors perform additional CV and charge-discharge measurements on pure PANI, coated on the Cu foil and/or on ICNs/PANI.

Response: Thanks for your comments. CV and charge-discharge measurements on ICNs/PANI and pure PANI have been performed in the revised manuscript( as shown in Figure S3, Figure S4 and Figure S5, supporting information).

  1. The capacity of ICNs/MoS2/PANI is rising after 500 cycles. How reproducible is this effect? In general, nowhere in the manuscript standard deviations or error bars are reported. How reproducible is the observed electrochemical behavior and the cyclic performance? How many cells were tested? This is especially crucial, when discussing quantitatively the specific materials properties. The authors should include also information and comment on the reproducibility of the presented preparation procedures.

Response: Thank you very much for this constructive comment. It has been confirmed that the capacity of ICNS/MoS2/PANI has a upward trend after 500 cycles, which is even maintained after 1000 cycles(Figure S6, supporting information). The reproducible experiment has been test for many times (no less than 10 cells), and the similar behavior also has been reported in previous literature (reference [18]). The increment of capacity maybe be attributing to the activation process of the hierarchical structure, the flower-like MoS2 gradually expanded and exfoliated during cycles, which can provide more lithium storage sites and low energy barrier for lithium intercalation or deintercalation. Further research is going on.

Reviewer 2 Report

A very fun job! very well designed. I only miss a more in-depth review of the literature, because there are few links. For example, Kurc et al in the work of Nanomaterials or Materials analyzes many electrode materials, it is worth quoting the literature of these authors and referring to them. MoS2 systems containing e.g. SiO2 are analyzed - it is worth comparing. After these small shortcomings are made up, the work can be published.

Author Response

A very fun job! very well designed. I only miss a more in-depth review of the literature, because there are few links. For example, Kurc et al in the work of Nanomaterials or Materials analyzes many electrode materials, it is worth quoting the literature of these authors and referring to them. MoS2 systems containing e.g. SiO2 are analyzed - it is worth comparing. After these small shortcomings are made up, the work can be published.

Response: Thank you very much for this constructive comment. Related references  (corresponding to references [12] and [28]) have been quoted in the revised manuscript.

Round 2

Reviewer 1 Report

The authors addressed properly all of my concerns and manage to improve the contribution.